Herbivore biocontrol and manual removal successfully reduce invasive macroalgae on coral reefs

Neilson Brian J. brian.j.neilson@hawaii.gov 1
Wall Christopher B. 2
Mancini Frank T. 2
Gewecke Catherine A. 1
1 State of Hawai‘i Division of Aquatic Resources , Honolulu , Hawai‘i , United States of America
2 Hawai‘i Institute of Marine Biology, University of Hawai‘i at Mānoa , Kāne‘ohe , Hawai‘i , United States of America
Reimer James
Electronic publication date: 2018 Aug 8
Publication date: 2018
Volume: 6
Electronic Location ID: e5332
Received 2018 Mar 26; Accepted 2018 Jul 6
Copyright: ©2018 Neilson et al.
Copyright year: 2018
Copyright holder: Neilson et al.
License: This is an open access article distributed under the terms of the Creative Commons Attribution License, which permits unrestricted use, distribution, reproduction and adaptation in any medium and for any purpose provided that it is properly attributed. For attribution, the original author(s), title, publication source (PeerJ) and either DOI or URL of the article must be cited.
License URL: https://creativecommons.org/licenses/by/4.0/

Keywords: Invasive species, Biocontrol, Macroalgae, Kaneohe bay, Kappaphycus, Eucheuma, Tripneustes, Gracilaria, Acanthophora, Coral reef

Funding: A Hawai‘i Invasive Species Council grant A NOAA Estuary Restoration Grant A USFWS Sport Fish Restoration Grant This work was made possible by funding provided by The State of Hawai‘i, Department of Land and Natural Resources, a Hawai‘i Invasive Species Council grant, a NOAA Estuary Restoration Grant, and a USFWS Sport Fish Restoration Grant. The funders had no role in study design, data collection and analysis, decision to publish, or preparation of the manuscript.

==============================
Invasive macroalgae pose a serious threat to coral reef biodiversity by monopolizing reef habitats, competing with native species, and directly overgrowing, and smothering reef corals. Several invasive macroalgae (Eucheuma clade E, Kappaphycus clade A and B, Gracilaria salicornia, and Acanthophora spicifera) are established within Kāne‘ohe Bay (O‘ahu, Hawai‘i, USA), and reducing invasive macroalgae cover is a coral reef conservation and management priority. Invasive macroalgae control techniques, however, are limited and few successful large-scale applications exist. Therefore, a two-tiered invasive macroalgae control approach was designed, where first, divers manually remove invasive macroalgae (Eucheuma and Kappaphycus) aided by an underwater vacuum system (“The Super Sucker”). Second, hatchery-raised juvenile sea urchins (Tripneustes gratilla), were outplanted to graze and control invasive macroalgae regrowth. To test the effectiveness of this approach in a natural reef ecosystem, four discrete patch reefs with high invasive macroalgae cover (15–26%) were selected, and macroalgae removal plus urchin biocontrol (treatment reefs, n = 2), or no treatment (control reefs, n = 2), was applied at the patch reef-scale. In applying the invasive macroalgae treatment, the control effort manually removed ∼19,000 kg of invasive macroalgae and ∼99,000 juvenile sea urchins were outplanted across to two patch reefs, totaling ∼24,000 m2 of reef area. Changes in benthic cover were monitored over 2 years (five sampling periods) before-and-after the treatment was applied. Over the study period, removal and biocontrol reduced invasive macroalgae cover by 85% at treatment reefs. Our results show manual removal in combination with hatchery raised urchin biocontrol to be an effective management approach in controlling invasive macroalgae at reef-wide spatial scales and temporal scales of months to years.

Introduction

Non-native macroalgae have been introduced worldwide (Schaffelke, Smith & Hewitt, 2006) as a result of spread through vectors including biofouling, ballast water, the aquarium trade, and seaweed mariculture (Ruiz et al., 2000; Zemke-White & Smith, 2006; Williams & Smith, 2007). Commercial macroalgae production has increased considerably in the last 50 years, becoming a multi-billion dollar industry in over 150 countries (FAO, 2015; Loureiro, Gachon & Rebours, 2015). Macroalgae mariculture occurs throughout tropical regions often cultivating non-native or domesticated species of the genera Caulerpa spp., Eucheuma spp., Gracilaria spp., and Kappaphycus spp. (Zemke-White & Smith, 2006; FAO, 2015; Radulovich et al., 2015). The macroalgae industry can provide economic opportunities for coastal communities and can offer a viable alternative to fisheries-based economies (Pickering & Forbes, 2002; Mate, Namudu & Lasi, 2003). However, macroalgae production can have inadvertent consequences for tropical reef biodiversity (Smith, Smith & Hunter, 2001; Stimson, Larned & Conklin, 2001; Ballesteros, 2006; Kružić, Žuljević & Nikolić, 2008; Longenecker, Bolick & Kawamoto, 2011; Martinez, Smith & Richmond, 2012; Sellers, Saltonstall & Davidson, 2015), contributing to a suite of anthropogenic pressures that are driving the global decline of live coral (Bruno & Selig, 2007; Gardner et al., 2003; Pandolfi et al., 2003; De’ath et al., 2012). Following the 2014–2015 global coral bleaching events (Van Hooidonk, Maynard & Planes, 2013; Eakin et al., 2017; Hughes et al., 2017) there is a need for immediate action to protect and restore coral reefs worldwide, including the management of invasive macroalgae on coral reefs.

Invasive macroalgae have the potential to negatively impact coral reefs by overgrowing reef building corals, outcompeting native species, and altering benthic habitat and the aquatic environment (i.e., chemistry, irradiance, sediment loading) (Russell, 1983; Woo, 2000; Conklin & Smith, 2005; Chandrasekaran et al., 2008; Rasher & Hay, 2010; Martinez, Smith & Richmond, 2012; Davidson et al., 2015; Sellers, Saltonstall & Davidson, 2015; Murphy & Richmond, 2016). Macroalgae contribute to ecosystem phase shifts from coral-dominated to macroalgae-dominated reefs (Done, 1992; Mumby et al., 2015; Dell et al., 2016; Van de Leemput et al., 2016). Such shifts to macroalgae dominance are generally associated with eutrophication, limited herbivory, or a combination of the two (Smith et al., 1981; Lapointe, 1997; Larned, 1998; Smith, Smith & Hunter, 2001; Stimson, Larned & Conklin, 2001; Thacker, Ginsberg & Paul, 2001; Vermeij et al., 2009). Phase shifts involving invasive macroalgae may pose additional competitive advantage over the native ecosystem. For instance, mariculture strains may have been selected for high growth and reproductive rates, vegetative propagation (Naylor, Williams & Strong, 2001; Ask & Azanza, 2002; Zhang et al., 2007), may be more tolerant of disturbed areas (Byers, 2002; Ruiz et al., 1999) and have limited preference by herbivores (Nyberg & Wallentinus, 2005; Boudouresque et al., 1996; Schaffelke, Evers & Walhorn, 1995). Considering the wide range of ecosystem services coral reefs provide (i.e., food security, tourism, shoreline protection, and cultural value) (Moburg & Folke, 1999), control and reduction of invasive macroalgae are a management priority for coral reef conservation.

Diverse techniques have been applied to eradicate or control marine macroalgae and include manual, chemical, and biological treatments (reviewed by Anderson, 2007). The type of technique applied depends on the management objective (i.e., eradication or control) and is often site and species specific (Anderson, 2007). Examples include chemical treatments (i.e., bleach, salt), thermal treatments (i.e., cold shock, heating), osmotic shock (i.e., freshwater and salinity treatments) (Cheshire et al., 2002; Williams & Smith, 2004; Wotton & Hewitt, 2004; Glasby, Cresse & Gibson, 2005; Anderson, 2007), mechanical or manual removal by hand and/or aided by vacuum or dredge pumps (Curiel et al., 2001; Miller et al., 2004; Hewitt et al., 2005; Conklin, 2007; Marks, Reed & Obaza, 2017), light attenuation, containment barriers, and water-removal with in situ desiccation (Anderson, 2007).

Biocontrol of invasive macroalgae is a newly emerging and promising macroalgae control technique. For instance, experimental use of sea urchins and mollusks in controlling invasive macroalgae species such as Caulerpa taxifolia, Caulerpa racemosa, and Codium fragile has been evaluated in the Mediterranean and Atlantic (Boudouresque et al., 1996; Thibaut & Meinesz, 2000; Scheibling & Hatcher, 2007; Cebrian et al., 2009). These studies revealed successful biocontrol applications have the highest impact in areas of low infestation (Scheibling & Hatcher, 2007; Cebrian et al., 2009) and suggest invertebrate biocontrols are most effective for emerging populations of invasive macroalgae. In some cases, the effectiveness of these treatments has been limited by macroalgae toxicity to biocontrol agents (Boudouresque et al., 1996), as well as the speed and the ability to produce and deploy adequate densities of biocontrol grazers to affected areas (Thibaut & Meinesz, 2000). Macroalgae abatement from herbivore biocontrol has recently shown promise on Hawai‘i’s reefs. The short-spined sea urchin, Tripneustes gratilla (Linnaeus) is a generalist herbivore native to Hawai‘i and will feed on at least five species of invasive macroalgae (Stimson, Cunha & Philippoff, 2007; Westbrook et al., 2015). T. gratilla has the potential for application as an invasive macroalgae biocontrol agent and has been shown to reduce macroalgae biomass within cage-enclosures in situ (Conklin & Smith, 2005; Stimson, Cunha & Philippoff, 2007; Chon, 2014; Westbrook et al., 2015). Moreover, T. gratilla has low mobility, can be easily handled, and maricultured from wild urchin stock and outplanted as juveniles (∼2.5 cm test diameter). Finally, T. gratilla achieves its maximum growth rate within the first two-years of life, and test size can reach 5.6–8.3 cm while grazing on invasive macroalgae species (Pan, 2012).

Invasive macroalgae are prominent in the Hawaiian archipelago. As a result, a number of aforementioned macroalgae control techniques have been tested in Hawai‘i (Smith et al., 2004; Conklin & Smith, 2005). Nineteen documented species of macroalgae have been introduced into Hawai‘i since the 1950’s, concentrated primarily on the island of O‘ahu where the main shipping and military ports are located (Russell, 1992; Smith, Hunter & Smith, 2002; DLNR, 2013). Several Rhodophyta macroalgae species have been particularly successful at invading Hawaiian reef communities, including Eucheuma clade E (N.L. Burman) F.S. Collins & Hervey, and Kappaphycus clade A and clade B (Doty) Doty ex P.C. Silva (Conklin, Kurihara & Shirwood, 2009), Acanthophora spicifera (Vahl) Børgesen, and Gracilaria salicornia (C. Agardh) E.Y. Dawson. The introduction of these macroalgae to Hawai‘i in the mid-20th century occurred through a variety of pathways including ship biofouling, ballast water discharge, and mariculture experimentation and production (Doty, 1961; Russell, 1983; Russell, 1992; Smith, Hunter & Smith, 2002).

Three Eucheumoid species of the genus Kappaphycus and Eucheuma from the Philippines, were intentionally planted on reefs around Moku o Lo‘e Island (Coconut Island) at the Hawai‘i Institute of Marine Biology (HIMB) (Kāne‘ohe Bay, Hawai‘i) for experimentation in the 1970’s (Doty, 1977; Russell, 1983). Molecular techniques (Zuccarello, Smith & West, 2006; Conklin, Kurihara & Shirwood, 2009) have identified these species as Kappaphycus clade A, Kappaphycus clade B, and Eucheuma clade E (hereafter Eucheuma). Prior to this analysis, nomenclature for these species has been inconsistent; therefore, we will refer to this group collectively as E/K hereafter unless referring specifically to species. E/K was left unchecked in Kāne‘ohe Bay for over two decades, and by 1996, E/K had spread >5 km from Moku o Lo‘e Island and were found throughout Kāne‘ohe Bay (Rodgers & Cox, 1999) and continued to spread to previously unaffected northern reefs adjacent to Kāne‘ohe by 1999 (Conklin & Smith, 2005). Eucheuma and Kappaphycus clade A are thought to spread only through vegetative propagation and their distribution has been restricted to Kāne‘ohe Bay, whereas Kappaphycus clade B is able to disperse vegetatively and sexually and has been documented outside of Kāne‘ohe Bay (Conklin, Kurihara & Shirwood, 2009). A. spicifera, the most widely distributed non-native macroalgae in Hawai‘i (Smith, Hunter & Smith, 2002), is thought to have been introduced and spread via ship biofouling or ballast water (Doty, 1961; Russell, 1983) or possibly through aquarium imports (Russell, 1992). A. spicifera is a common fouling species on ship hulls and is able to disperse sexually and via vegetative fragmentation, which may explain its wide distribution (Smith, Hunter & Smith, 2002). The origin of G. salicornia are speculative, possibly arriving to Hilo Bay in the 1940’s associated with ships originating from the Philippines (Smith et al., 2004) and then later intentionally transplanted to various sites around Moloka‘i and O‘ahu, including Kāne‘ohe Bay (Russell, 1992; Smith, Hunter & Smith, 2002; Smith et al., 2004). G. salicornia is thought to disperse primarily via vegetative fragmentation (Smith et al., 2004).

All five species are capable of forming dense mats on the reef, overgrowing reef corals, and monopolizing reef habitats (Russell, 1983; Ask & Azanza, 2002; Conklin & Smith, 2005; Martinez, Smith & Richmond, 2012). E/K has been shown to be particularly damaging to corals by shading and smothering live coral and can eventually lead to mortality (Russell, 1983; Woo, 2000; Conklin & Smith, 2005; Chandrasekaran et al., 2008). G. salicornia can also impact reef corals by decreasing irradiance via smothering, altering water chemistry (i.e., hypoxia and hypercapnia) and increasing sedimentation surrounding reef corals (Martinez, Smith & Richmond, 2012). Although five of these invasive macroalgae species are thought to be damaging to reef biodiversity, E/K were deemed a management priority due to its especially damaging impacts to corals and its limited distribution compared to A. spicifera and G. salicornia (DLNR, 2013).

In response to the destructive impact to corals and the concern that E/K would continue to spread and establish on reefs beyond Kāne‘ohe Bay, local managers, community members, and researchers worked to develop a control technique for invasive macroalgae with particular focus on E/K. Conklin & Smith (2005) tested various control methods and found that E/K quickly regrew after manual removal, but sea urchin biocontrol showed a sustained reduction of E/K in small-scale field trials. Conklin & Smith (2005) recommended combining techniques by using manual removal to reduce the bulk of E/K biomass, followed by sea urchins biocontrol treatment to reduce re-growth. Preliminary field trials conducted by Hawai‘i Department of Land and Natural Resources on a patch reef in Kāne‘ohe Bay supported this observation (DLNR, 2013). Based on these findings and recommendations, a large-scale invasive macroalgae control project on patch reefs in Kāne‘ohe Bay was initiated in 2008 using the combination of manual removal and sea urchin biocontrol.

The overarching goal of the project was the rehabilitation and preservation of coral reef habitat and associated biodiversity with specific management objectives to: (i) reduce invasive macroalgae on Kāne‘ohe Bay patch reefs, and (ii) stop the spread of E/K to unaffected reefs within and outside Kāne‘ohe Bay. Although the macroalgae control techniques applied in this study were evaluated previously in small-scale experiments, the combined use of manual removal and sea urchin biocontrol has yet to be tested as a management-relevant, reef-wide scale approach. In this study we evaluate the effectiveness of manual removal combined with urchin biocontrol in sustaining a reduced invasive macroalgae cover (E/K (i.e., Eucheuma, Kappaphycus clade A, Kappaphycus clade B), G. salicornia, A. spicifera) at a reef-wide scale over 2 years using a Before After Control Impact (BACI) experimental design. We hypothesized that our proposed invasive macroalgae removal and control methods would be effective at maintaining low invasive macroalgae abundance (percent cover) over time at treatment reefs relative to untreated-control reefs. While, a factorial design testing each treatment type separately (i.e., manual removal, biocontrol, and combined treatments) might be preferred, this, was not possible due to logistic and financial challenges associated with implementing and replicating three separate treatment types at the reef-wide scale. However, previous findings of Conklin & Smith (2005) and data from the State of Hawai‘i Division of Aquatic Resources at a scale smaller than the one applied in the current study showed manual removal of invasive algae in the absence of biocontrol cannot successfully reduce invasive macroalgae cover over long term. Simply applying urchin biocontrol without manual removal was also not advised based on concerns of increased fragmentation by urchins detaching holdfasts of large E/K mats. In addition, applying urchins to a large standing crop of macroalgae would increase the amount of urchins, grazing time, and ultimate cost required to successfully treat a reef. Therefore, our goal was to use a single, most-effective treatment type (i.e., the combination of manual removal and biocontrol) and test whether this treatment was effective at reducing invasive algae cover long term among replicate patch reefs.

Materials and Methods

Study site

Invasive macroalgae removal and biocontrol techniques were carried out on four shallow (0.5–2.0 m depth) patch reefs located in central Kāne‘ohe Bay, on the windward side of O‘ahu, Hawai‘i (21°28′0″N, 157°49′0″W), which is the largest embayment in the Hawaiian Islands and contains over 70 distinct patch reefs surrounded by a barrier reef and fringing reef system (Fig. 1). The patch reefs are island-like features separated by 10–15 m sand bottom. Two patch reefs (Reef 26 and 27) were designated as treatment reefs, where manual removal of E/K and sea urchin biocontrol were applied, and two patch reefs (Reef 16 and 28) were designated as control reefs where no macroalgae manual removal or biocontrol were applied (Fig. 1). Study reefs were selected based on the presence of invasive macroalgae and their close proximity to each other. Designated patch reefs were approximately 11,900 m2 (treatment Reef 26), 12,700 m2 (treatment Reef 27), 3,100 m2 (control Reef 16), and 14,500 m2 (control Reef 28). Each patch reef has a distinct reef slope composed primarily of live coral and a shallower reef flat consisting of a mix of live coral, dead coral, rubble, and sand. E/K occurred on reef slopes and reef flats and ranged in size from single low growing thalli to dense mats 1 m2 in area and ∼0.3 m thick (Figs. 2A–2B). G. salicornia and A. spicifera occurred primarily on the reef flats and also ranged from single thalli to mats >1 m2 and ∼0.1 m thick (Figs. 2C–2D).

Figure 1 Study site location in Kāne‘ohe Bay on the windward side of the island of O‘ahu, Hawai‘i, proximate to Moku o Lo‘e (Hawai‘i Institute of Marine Biology).

Baseline image provided by ©DigitalGlobe, Inc., All Rights Reserved.

Figure 2 Invasive macroalgae species found on study reefs in Kāne‘ohe Bay.

(A) Eucheuma clade E, (B) Kappaphycus clade B, (C) G. salicornia, (D) A. spicifera (photo credit: Brian Neilson).

Figure 3 Invasive macroalgae control techniques applied in the field.

(A) using the Super Sucker to manually remove Eucheuma clade E, (B) outplanting juvenile T. gratilla, (C) outplanted adult T. gratilla surrounded by G. salicornia and A. spicifera, (D) adult T. gratilla surrounded by Eucheuma clade E, (E) before and immediately (F) after manual removal of Eucheuma clade E revealing crustose coralline algae (CCA) and (G) before and (H) after removal of Eucheuma clade E revealing live and dead coral (photo credit: (A–B) DLNR/DAR, (C–H) Brian Neilson).

Invasive macroalgae control technique

Invasive macroalgae were controlled in two phases. First, E/K were manually removed from reefs by divers aided by an underwater vacuum system (“The Super Sucker”) that transported macroalgae from the reef to a support vessel (Fig. 3A) (Conklin, 2007). To a lesser extent, divers manually removed and bagged macroalgae without aid of the Super Sucker system. At the support vessel, macroalgae was bagged, weighed (wet weight to the nearest kg), and then delivered to farmers in the Kāne‘ohe Bay watershed for use as an agricultural fertilizer. Manual removal was conducted from November 2011 to March 2012 on treatment Reef 26 over 23 working days and treatment Reef 27 was cleared from March 2012 to August 2012 over 25 working days (Table 1). Divers removed the bulk of the E/K biomass, leaving macroalgae in hard-to-reach areas (e.g., between coral branches and within crevices), small clumps (<400 cm3) and holdfasts to maximize the yield to effort ratio and minimize disturbance to other benthic organisms and habitats. Invasive macroalgae species G. salicornia and A. spicifera were not directly targeted by divers for manual removal.

Table 1 Invasive macroalgae manual removal and T. gratilla outplanting dates, area, and stocking density.

Treatment reef	Manual removal dates	Manual removal days	E/K removed (kg)	Urchin outplanting dates	Urchin outplanting area (m2)	Urchins stocked	Urchin stocking density (urchinsm−2)	
Reef 26	Nov 2011–Mar 2012	23	11,963	Dec 2011–Dec 2013	11,900	46,913	3.94	
Reef 27	Mar 2012–Aug-2012	25	7,095	Aug 2012–Dec 2013	12,700	52,835	4.16	
Total	Nov 2011–Aug-2012	48	19,058	Dec 2011–Dec 2013	24,600	99,748	4.05	

Sea urchin biocontrol

Adult T. gratilla were collected from the wild and spawned at an urchin hatchery. Urchin larvae were settled and reared in tanks on land until they reached approximately 2.5 cm diameter test size (∼4–6 months after spawning). A new cohort was produced every 30–60 days throughout the duration of the study. Following E/K manual removal, juvenile urchins were transported to the reef in trays and manually deployed on the treatment reefs where G. salicornia, A. spicifera, and E/K occurred (Figs. 3B–3D). A systematic approach was used to deploy urchins to achieve a relatively consistent urchin density throughout the entire reef. Urchins were deployed to the reef as they became available by the hatchery, requiring repeated stocking events to treat each reef. Additional urchins were spot-treated to areas that remained high in invasive macroalgae cover and/or void of urchins as a result of attrition or being inadvertently missed during the initial deployments. Hatchery related biosecurity protocols were followed to prevent the spread of disease and invasive species, and urchins were closely monitored for signs of disease or abnormalities.

On treatment Reef 26, a total of 46,913 T. gratilla were outplanted to affected areas, the majority of which (76% of total) were outplanted from December 2011 to October 2012, with supplemental outplanting from July to December 2013 (19%) one additional outplanting in July of 2014 (13% and 5% of total, respectively) (Table 1). On treatment Reef 27, a total of 52,253 urchins were outplanted (Table 1), primarily from August 2012 to May 2013 (97% of total) with one additional supplemental stocking (1,500 urchins) in December 2013. Stocking density of juvenile urchins was 3.9 urchins m−2 on treatment Reef 26 and 4.2 urchins m−2 on treatment Reef 27 (Table 1).

Invasive macroalgae control costs

Control costs were calculated for field operations (i.e., manual removal and sea urchin outplanting) and sea urchin hatchery operations. Cost estimates included salaries and operating expenses (i.e., equipment, materials, supplies, fuel, and utilities). The total invasive macroalgae control cost of the project was divided by the total reef area treated to estimate cost per square meter.

Benthic surveys

Baseline benthic surveys were performed at all patch reefs from November 2011 to February 2012 (hereafter, Winter 2011) prior to macroalgal removal and urchin outplanting, representing the “before” period of the analysis. Subsequently, benthic surveys were repeated during the treatment period at four additional times during summer and winter seasons from 2012–2014, representing the “after” periods of the analysis. Sampling periods were defined as: May–June 2012 (hereafter, Summer 2012), December 2012–February 2013 (hereafter, Winter 2012), May–June 2013 (hereafter, Summer 2013), and February 2014 (hereafter, Winter 2013). Using these five time points we analyzed changes in percent cover of invasive macroalgae (Eucheuma, Kappaphycus clade B, G. salicornia, A. spicifera), native macroalgae, coral, crustose coralline algae (CCA), and the combined sand/rubble, bare space, turf (thallus length ≤10 mm) (SBT) at treatment and control reefs.

Fixed transect locations were randomly selected initially using ArcGIS random point tool (ESRI, 2011) within the following strata: windward and leeward prevailing wind orientation (northeast) and habitat type (aggregate reef, mixed/unconsolidated reef, and pavement/consolidated reef situated on reef flat and reef slope areas). A windward/leeward stratification was applied to control for the possibility of detached algae collecting disproportionately on the leeward side of reefs as a result of wind driven currents. Reef flat transects ran perpendicular to the prevailing wind direction at a bearing of ∼140°. Reef slope transects followed the ∼1 m depth contour clockwise around the reef. The number of transects per reef varied according to reef size at an average sampling effort of one transect per ∼800 m2. The total fixed transects for each reef were: 6 (control Reef 16), 18 (control Reef 28), 13 (treatment Reef 26), and 14 (treatment Reef 27). The number of transects were allocated in proportion to the total reef area first, then by primary reef habitats (aggregate and non-aggregate), then by non-aggregate sub-strata (mixed/unconsolidated reef and pavement/consolidated reef). Mean benthic cover was estimated using a point intercept transect method (Hill & Wilkinson, 2004). Surveyors recorded the benthic cover at 0.2 m intervals along a 25 m transect (n = 126 points transect−1). T. gratilla were surveyed at each transect location, counting all observed individuals within a 25 × 1 m belt. A correction factor of 90% detectability (based on F Mancini and D Minton field trials) was used to estimate the density of urchins from transect counts.

Data analysis

Response variables (percent cover of invasive macroalgae, native macroalgae, CCA, coral and SBT) were monitored over time, with baseline surveys at each patch reef (Winter 2011) designated as the “before” period and four subsequent surveys (Summer 2012, Winter 2012, Summer 2013, Winter 2013) designated as “after” periods. Treatment application (i.e., algae removal plus urchin outplanting) was partial in Summer 2012 and complete by Winter 2012 (Table 1). Changes in community cover were assessed using a linear mixed effects model fit by restricted maximum likelihood in the lme4 package (Bates et al., 2015) in R version 3.3.0 (R Development Core Team, 2017). Treatment (E/K manual removal and biocontrol vs. no E/K removal or biocontrol) and time (before treatment applied vs. periods after treatment applied) were included as fixed effects. To account for spatial structure of the benthos habitat types within patch reefs (i.e., aggregate reef, mixed/unconsolidated reef, and pavement/consolidated reef) habitat was designated as a random effect nested within individual reefs. Reef transects were included as a repeated-measure random effect. Considering that surveys conducted over the two-year study period spanning different months and seasons, we first tested ‘season’ (i.e., summer vs. winter) separately as a fixed effect in a linear model; no effects were observed (p ≥ 0.408) and season was not included in the final analysis. Normality of residuals and homogeneity of variance was verified using graphical inspection of standardized residuals, and transformations were applied where assumptions of ANOVA were not met. An arcsine transformation was used for invasive algae and abiotic cover and a square root transformation was used for CCA and native macroalgae. Analysis of variance tables were generated using type-II sum of squares with Satterthwaite approximations of degrees of freedom using the package lmerTest (Kuznetsova, Brockhoff & Christensen, 2017). Where significant interactions were found, posthoc slice tests were performed using lsmeans (Lenth, 2016) to evaluate differences between control and treatment reefs within each sampling time point. All data and code to reproduce figures and analyses can be found on Zenodo (10.5281/zenodo.1285551).

Results

Initial field surveys

Mean benthic cover was comparable for all groups (i.e., invasive and native algae, coral, CCA, bare substrate) (posthoc: p ≥ 0.721) at treatment and control reefs at the start of the study (Figs. 4A–4E). In Winter 2011, benthic cover at the four study reefs was, on average, dominated by hard corals (mean ± SE) (39 ± 13%), followed by invasive macroalgae (21 ± 5%), CCA (5 ± 2%), and native macroalgae (5 ± 2%). The native macroalgae community cover was composed primarily of Dictyosphaeria versusii (74 %) and Dictyosphaeria cavernosa (19%). Invasive macroalgae on control reefs was predominantly G. salicornia (11%), and Eucheuma (7%), whereas invasive macroalgae cover at treatment reefs had similar cover of G. salicornia, A. spicifera, and Eucheuma (5–7%) (Figs. 5A–5B). Kappaphycus clade B made up the smallest component of the invasive macroalgae community (0–2.5%) on all study reefs (Figs. 5A–5B). T. gratilla was not detected on control or treatment reefs in the pre-treatment surveys.

Figure 4 Mean percent cover of benthic cover types.

(A) combined invasive macroalgae (Eucheuma clade E/Kappaphycus Clade B/Acanthophora spicifera/Gracilaria salicornia), (B) native macroalgae, (C) crustose coralline algae (CCA), (D) corals, and (E) SBT (sand/bare/turf). Values are mean ± SE; n = 24 (control) and n = 26–27 (treatment) for each sampling time. The first time point in each figure (Winter 2011) represents the “before” time period of the study and all subsequent time points represent the “after” period. Symbols (*) represent a significant difference (p ≤ 0.05) between the control and treatment.

Figure 5 Percent cover for invasive macroalgae species through time.

(A) control reefs and (B) treatment reefs. Values are mean ± SE; n = 24 (control reefs) and n = 26–27 (treatment reefs) for each sampling time.

Macroalgae removal and urchin outplanting surveys

Divers removed a total of 11,963 kg wet weight (0.81 ± 0.14 kg m−2) of E/K from different areas of treatment Reef 26 and 7,095 kg wet weight (0.622 ± 0.05 kg m−2) from treatment Reef 27 (Table 1). The majority of macroalgae was removed using the Super Sucker (80%) versus hand removal using bags (20%). E/K was cleared at an average rate of 1.48 ± 0.14 m2 min−1. On treatment reefs, the mean (±SE) removal effort was greater for Reef 26 (2.36 ± 0.27 m2 min−1) compared to treatment Reef 27 (1.23 ± 0.10 m2 min−1) as well as the E/K biomass removed 0.81 ± 0.14 kg m−2(Reef 26) versus 0.62 ± 0.05 kg m−2 (Reef 27). While stocking density of juvenile urchins was designed to be ∼4 urchins m−2 (Table 1), field surveys following urchin outplanting estimated urchin densities was 0.90 urchins m−2 (treatment Reef 26) and 0.74 urchins m−2 (treatment Reef 27). No presence of T. gratilla was reported in benthic surveys on control Reefs 16 and 28 in post treatment surveys.

Post-macroalgae removal and urchin outplanting surveys

E/K macroalgae manual removal and urchin biocontrol led to an 85% decline in invasive macroalgae cover over the study period, from 21% cover in Winter 2011 to 4% cover in Winter 2013 (Figs. 4A, 5B, 6). Invasive macroalgae cover was affected by the interaction between treatment and time (Table 2). On treatment reefs, percent cover of Eucheuma– a target of manual macroalgae removal–had declined by 59% at the first sampling time (Summer 2012), approximately 6 months after the treatment had been applied (Fig. 5B). However, total invasive macroalgae cover on treatment reefs did not significantly differ from control reefs until one year after the treatment application had begun (posthoc: p = 0.029). By Winter 2012 total invasive macroalgae cover had declined by 29% relative to Winter 2011 levels. The mean invasive macroalgae cover at control reefs fluctuated between 14–25% over the entire study period (Winter 2011 to Winter 2013) (Figs. 4A, 6) and comparable across all time points, (posthoc: p ≥ 0.080) except Winter 2013 where invasive algae declined relative to start of the study (posthoc: p = 0.005). G. salicornia and Eucheuma consistently dominated the invasive macroalgae community at control reefs (Fig. 5A), representing mean cover of 7–12% at each sampling time throughout the study period.

Figure 6 Mean percent cover for benthic community members at control and treatment reefs before applying treatments (Winter 2011) and two years after treatment application (Winter 2013).

Values are mean ± SE; n = 24 (control) and n = 26–27 (treatment) for each sampling time.

Table 2 Analysis of variance table for treatment and time effects on coral reef community cover.

Dependent variable	Effect	SS	df	F	p	
Invasive macroalgae	Treatment	0.031	1, 9	3.377	0.098	
	Time	1.478	4, 195	40.389	<0.001	
	Treatment × Time	0.629	4, 195	17.202	<0.001	
Native macroalgae	Treatment	0.0005	1, 9	0.015	0.906	
	Time	0.120	4, 195	8.841	<0.001	
	Treatment × Time	0.026	4, 195	1.928	0.107	
CCA	Treatment	0.0005	1, 9	0.045	0.837	
	Time	0.366	4, 195	9.194	<0.001	
	Treatment × Time	0.104	4, 195	2.606	0.037	
Coral	Treatment	0.0001	1, 9	0.056	0.818	
	Time	0.181	4, 195	34.783	<0.001	
	Treatment × Time	0.020	4, 195	3.867	0.005	
Sand/bare/turf	Treatment	0.007	1, 9	0.520	0.489	
	Time	0.072	4, 195	1.344	0.255	
	Treatment × Time	0.048	4, 195	0.893	0.469	
Notes.

Linear mixed effect models fit by restricted maximum likelihood; analysis of variance table of Type II sum of squares and Satterthwaite approximation for degrees of freedom.

Invasive macroalgae Eucheuma denticulatum, Kappaphycus alvarezii, Acanthophora spicifera, Gracilaria salicornia

CCA crustose coralline algae

SS sum of squares

df degrees of freedom in numerator and denominator

Bold p values represent significant effects (p < 0.05).

Mean native macroalgae percent cover ranged from 2–5% over the study period and decreased over time (p < 0.001) but not in response to treatments (p = 0.906) (Table 2) (Fig. 4B). The interaction of treatment × time affected coral (p < 0.001) and CCA cover (p = 0.037), and both coral and CCA increased over the study period (p < 0.001). However, mean coral and CCA cover did not differ among control and treatment reefs at each discrete time point (posthoc: p ≥ 0.286). SBT (sand/bare/turf) was not affected by time, treatment, or their interaction (p ≥ 0.255) (Table 2), but tended to be lower at control reefs (25–30% cover) relative to treatment reefs (35–40% cover) (Fig. 4E).

Invasive macroalgae control costs

Field components of macroalgae removal and control operations cost an estimated $255,000 and roughly 3,000 human hours of work. Hatchery operations cost $562,000, and accounted for approximately 19,000 human hours to run the hatchery facility, which required daily oversight. The total project cost $817,000 to treat 24,600 km2 ($33 m−2) of affected reef.

Discussion

Effectiveness of invasive macroalgae control

For invasive macroalgae control, there are few demonstrated techniques available for managers when prevention and eradication attempts have failed and valuable resources and biodiversity are at risk (Anderson, 2007). Further, there are few examples of macroalgae control techniques being successfully applied beyond small-scale experiments. The present study demonstrates manual removal of invasive macroalgae, in combination with outplanting hatchery raised juvenile urchins (T. gratilla) for biocontrol, can be an effective approach for reducing the benthic cover of invasive macroalgae at a reef-wide scale. Invasive macroalgae was reduced by 85%, two-years after macroalgae removal and sea urchin biocontrol was applied—a result consistent with a small-scale experiment that employed a similar control technique over a shorter time period (Conklin & Smith, 2005).

The treatments applied in this study showed promising results in controlling invasive macroalgae. Manual removal aided by the Super Sucker system was an effective means to remove E/K biomass (51% decline post manual removal) and was also an efficient means of moving thousands of kilograms of macroalgae from the reef to the support vessel at a mean removal rate of 1.48 ± 0.14 m2 min−1. In addition, the vacuum system captured loose macroalgae fragments created by dislodging the macroalgae, reducing the risk of unintentional dispersal. Following manual removal, invasive macroalgae continued to decline by 61% from Winter 2012 to Winter 2013 (Fig. 4A). Although individual treatment types were not tested here, we speculate that this decline was a result of T. gratilla biocontrol based on the findings of Conklin & Smith (2005), which documented steady re-growth of E/K without T. gratilla biocontrol. It should be noted that manual removal and sea urchin biocontrol manipulations deployed in this study took several months to carry-out (Table 1) and supplemental urchins were added to reefs throughout the study to account for attrition. Therefore, the first “after period” (i.e., Summer 2012) may be viewed as a transitional period in the chronology of our experiment, bridging pre-manipulation and full treatment establishment periods.

Assessing invasive algae mitigation at reef-wide scales has a strong and direct application to management, however, such studies also present challenges in terms of replication and sample size. Alternatively, studies conducted at smaller spatial scales (i.e., plot-level) offer greater replication, but results may not necessarily be extrapolated to larger scales. In regards to this study, treatments were not fully crossed and replication was low, however, our method are promising, especially in demonstrating the potential for a native, mobile invertebrate as an effective biontrol agent. Despite shortcomings, our results show a clear and lasting results of reduced invasive macroalgae on treatment reefs, indicating our approach was successful and effective in controlling invasive macroalgae over 24,600 km2 of coral reef habitat

The sea urchin, T. gratilla, are well suited for mariculture and outplanting for the biocontrol of invasive macroalgae. T. gratilla are able to be propagated in a hatchery using wild stock, producing large numbers of juvenile urchins (∼100,000 yr−1) (DL Cohen, pers. comm., 2017) without impacting wild T. gratilla populations. The small size (∼2.5 cm test diameter) of outplanted T. gratilla may also be an important factor in treating invasive macroalgae. Chon (2014) found small urchins (0.5–2.5 cm test diameter) were more effective at grazing invasive macroalgae than adult T. gratilla (∼4 cm test diameter) within in situ enclosures. Thus, small test-size urchins appear more capable of grazing holdfasts within the small interstitial spaces of the reef. As juvenile urchins mature, they continue to contribute to invasive macroalgae biocontrol (Chon, 2014; Westbrook et al., 2015), but possibly to a lesser extent. Therefore, the potential for T. gratilla as a biocontrol agent may be size-dependent (Chon, 2014).

While the primary target species for manual removal was E/K, other invasive macroalgae not targeted by manual removal (G. salicornia and A. spicifera) also declined over the study period (Fig. 5B). Potentially, the reductions in G. salicornia and A. spicifera cover at treatment reefs are due to urchin herbivory reducing the cover of these non-targeted (for manual removal) invasive macroalgae. In feeding trials, T. gratilla consumed all four species of invasive macroalgae found in this study, but given the choice, urchins preferred A. spicifera, especially among smaller test-size urchins (Westbrook et al., 2015). T. gratilla will also graze G. salicornia, but displays the least preference for this species (Stimson, Cunha & Philippoff, 2007; Westbrook et al., 2015). Further, Westbrook et al. (2015) found that T. gratilla were able to graze invasive macroalgae at a rate of 7.5 g d−1 per urchin, which they estimated to be roughly equal to the growth rate of the four species of invasive macroalgae examined.

This study demonstrates that T. gratilla biocontrol can be successful when applied at the scale of a patch reef (∼12,000 m2). However, since urchin movement was naturally confined by 10–15 m deep sandy habitats surrounding patch reefs in Kāne‘ohe Bay, this raises the question as to whether T. gratilla would be as effective in treating larger continuous reefs. Valentine & Edgar (2010) detected a significant decline of macroalgae on continuous reef habitats when T. gratilla are present in high densities (>4 m−2) at Lord Howe Island. Stimson, Larned & Conklin (2001) found T. gratilla movement to be <1 m d−1 and suggested that this low vagility may explain its generalist diet of a wide range of macroalgae species including non-natives. The low movement rates have allowed Hawai‘i managers to utilize T. gratilla in spot-treating areas with high invasive macroalgae biomass and apply a manipulated urchin density in problematic locales. Therfore, T. gratilla shows promise as a macroalgae biocontrol agent, but assessing its function in different reef systems should be a priority for future research.

It is reasonable to acknowledge the potential risk of urchin stocking in Kāne‘ohe Bay to facilitate rapid T. gratilla population growth. However, we believe this risk is unlikely due to a number of factors, including T. gratilla stocking densities were similar to those observed in natural populations on Hawaiian reefs (Walsh et al., 2012). In addition, outplanted urchins remain under pressure from a wide range of natural predators such as fish, decapods, and cephalopods, and urchins were closely monitored by resource managers. Although urchins are reproductively viable, for reasons unknown, conditions in Kāne‘ohe Bay have not been favorable for T. gratilla recruitment to patch reefs, and no natural recruitment of hatchery raised T. gratilla in Kāne‘ohe Bay has been observed (B Neilson, 2017, unpublished data).

Invasive macroalgae declined across all four reefs examined in the study, which may have been related to environmental factors such as nutrients, water motion, temperature, and salinity (Glenn & Doty, 1990) throughout Kāne‘ohe Bay. Herbivorous reef fish grazing has also been demonstrated to have a profound impact on macroalgae cover (Williams & Polunin, 2001; Burkepile & Hay, 2006; Hughes et al., 2007; Rasher, Hoey & Hay, 2013) and may have also contributed to macroalgae decline. For instance, Stamoulis et al. (2017) found G. salicornia was the second most prevalent macroalgae species in gut contents of herbivorous reef fishes in Kāne‘ohe Bay. While E/K and A. spicifera were also identified in fishes gut contents, these species were far less prevalent (Stamoulis et al., 2017). Although herbivorous fishes appear to be a substantial contributor to controlling invasive macroalgae, protection of herbivorous fishes (in a small marine protected area) alone was not able to reduce invasive macroalgae levels significantly in all reef habitats (Stamoulis et al., 2017). Other Hawaiian reefs that have protection rules in place for herbivores, including T. gratilla, have found significant reductions in macroalgae including A. spicifera (Williams et al., 2016). Based on the findings of this study and others (Conklin & Smith, 2005; Stimson, Cunha & Philippoff, 2007; Westbrook et al., 2015; Chon, 2014), T. gratilla appears to be the most effective single biocontrol species when combined with manual removal for treating invasive macroalgae on Hawai‘i coral reefs.

T. gratilla are effective invasive macroalgae grazers (Conklin & Smith, 2005; Stimson, Cunha & Philippoff, 2007; Chon, 2014; Westbrook et al., 2015), however, it has been suggested that urchin herbivory may have negative effects. For instance, indiscriminate low-profile grazing on the reef substratum may reduce the survival of juvenile corals (Forsman, Rinkevich & Hunter, 2006), newly settled coral recruits, or CCA (Stimson, Cunha & Philippoff, 2007). CCA are an important component of reef structure and stability (Bak, 1976), in addition to providing a substratum for coral recruitment and development (Morse et al., 1996; Negri et al., 2001; Harrington et al., 2004). However, we found no negative effects of treatments (i.e., manual removal + urchin grazing) on coral cover or CCA. Instead, coral cover and CCA showed positive trends through time independent of treatments. Similarly, Stanley (2014) found T. gratilla had no effect on settlement or survival of six Kāne‘ohe Bay coral species and Valentine & Edgar (2010) found T. gratilla outbreaks had no effect on coral cover in Lord Howe Island, Australia. Together, these results suggest T. gratilla stocked at densities for biocontrol actions do not appear detrimental to reef corals or ecologically important CCA. Although no treatment effect of native macroalgae was observed in this study, we speculate that urchin biocontrol may inhibit native macroalgae growth and colonization based on the findings of Valentine & Edgar (2010) following a T. gratilla outbreak and Chon (2014) who found a significant decline in native macroalgae in T. gratilla enclosure experiments at stocking density >2 urchins m−2. Although the density of urchins in this study (0.74–0.9 urchins m−2) was lower than Chon’s (2014) recommended 2 urchins m−2, it still may be advisable to reduce urchin densities once urchins have grazed invasive macroalgae to <2% cover in order to limit potential negative effects on native macroalgae colonization and growth.

The observed decrease in invasive macroalgae on control reefs over the course of the study did not result in a significant increase in any single benthic cover type as a result of the treatment (Fig. 4). However, the benthic community composition appears to have changed throughout the course of the study (Fig. 6). This shift from areas dominated by invasive macroalgae to a mix of coral, CCA, native macroalgae, and SBT (sand/bare/turf) may favor the settlement of native flora and fauna and increase the accessibility of suitable settlement substratum. Additionally, the application of manual removal plus urchin biocontrol resulted in no reductions in ecologically important benthic groups, such as corals and CCA. Approaches to control invasive macroalgae are diverse and not always benign (Anderson, 2007), and applying such treatments on ecologically sensitive habitats, such as coral reefs, demand minimal environmental impacts. Although E/K was carefully hand removed from the reef and fed into the vacuum system, the process does cause low levels of disturbance to the reef including abrasion to coral tissue, dislodgment of benthic organisms and habitat, and the potential for bycatch of small or cryptic organisms associated with macroalgae. It is therefore advised that macroalgae manual removal be performed once or infrequently to reduce the potential for environmental disturbance.

Manual removal of invasive macroalgae in combination with sea urchin outplanting proved to be a successful approach for invasive macroalgae mitigation in Hawai‘i. However, the substantial costs and labor requirements, as well as the necessity of a native herbivore amenable to culturing and outplanting, may limit the broad application of this approach in other reefs or at broader scales than those tested here. Therefore, the effectiveness of this approach on other reef systems required appropriate testing at small experimental scales before reef-wide treatments are applied (Conklin & Smith, 2005; DLNR, 2013), in addition to long term financial support to advance laboratory tests to the reef-wide scale. Such tests are necessary to evaluate environmental impacts, the need for manual removal, sea urchin biocontrol or both in controlling invasive macroalgae, and weighing logistic and financial constraints.

Control costs

The control of invasive macroalgae for this study was a substantial investment by managers at a cost of $817,000 to treat 24,600 km2 ($33 m−2) of affected reef. This figure not only demonstrates the need to invest in invasive species prevention through strict vector management and importation rules, but also indicates the importance of Hawai‘i’s reefs in order to justify such a large expense. Cesar & Van Beukering (2004) estimated a 360 million dollar a year net benefit for Hawai‘i’s economy and a total value of 10 billion dollars. Therefore, investment in restoration and preservation of coral reef ecosystems by controlling invasive macroalgae may be a worthwhile economic investment. It should also be noted that the cost per m2 of treated reef can be reduced by further advances in sea urchin aquaculture.

Conclusion

Our findings show that manual removal and sea urchin biocontrol applied at a reef-wide scale is an effective approach for controlling invasive macroalgae, but should not be viewed as a replacement for managing some of the other drivers of macroalgae phase shifts, including increased nutrients (Lapointe, 1997; Stimson, Larned & McDermid, 1996), and reduced herbivory (Hay, 1984; Hughes, 1994; Larned, 1998; Bellwood et al., 2004). In addition, the long-term effects (>3 years) are unknown and will require further monitoring in the years to come. The control techniques demonstrated in this study combined with watershed (Richmond et al., 2007) and herbivore (Mumby & Steneck, 2008) management are necessary to achieve broad goals of reef restoration and habitat improvement. Marine reserves and Herbivore Fisheries Management Areas have shown positive results in Hawai‘i, by increasing biomass of herbivorous reef fish and reducing cover of invasive macroalgae (Friedlander, Brown & Monaco, 2007; Williams et al., 2016). Unfortunately, native reef fish and urchin assemblages may not be capable of controlling the combined growth rates of multiple invasive macroalgae species, and therefore, a suite of management strategies may be necessary to control invasive macroalgae at a large-scale.

Supplemental Information

Data S1 Raw data

Click here for additional data file.

Supplemental Information 1 R markdown file

Click here for additional data file.

Supplemental Information 2 HTML output of R markdown

Click here for additional data file.

We would like to acknowledge the contribution to this project by the following individuals and organizations: DAR Staff: Jonathan Blodgett, Karen Brittain, Vince Calabrese, David Cohen, Megan Cook, Michael Fujimoto, Justin Goggins, Katharine Hind, Brian Kanenaka, Dan Lager, Derek Levault, Sean Louie, Paul Murakawa, Kimberly Peyton, Andrew Purves, Neil Rodriguez, Brad Stubbs, Kendall Tejchma, Travis Thyberg, Jackie Troller, Ray Uchimura, and Tristen Walker. The Hawai‘i Coral Reef Initiative/Social Science Research Institute: Kristine Davidson, Pamela Fujii, Mike Hamnett, and Charissa Minato. The Nature Conservancy: Ryan Carr, Eric Conklin, Justin Dennis, Jan Eber, Kim Hum, Hank Lynch, Dwayne Minton, and Kanekoa Shultz, Eva Zafarano, Kāne‘ohe Bay Farmers: Charlie Reppun, John Reppun, and Paul Reppun. Hawai‘i Institute of Marine Biology and University of Hawai‘i: Megan Donahue, Mary Donovan, Kyle Edwards, Zac Forsman, Erik Franklin, Ku‘ulei Rodgers, Kosta Stamoulis, John Stimson, Rob Toonen, and Charley Westbrook. NOAA: Mathew Parry, USFWS: Tony Montgomery, The Pacific Cooperative Studies Unit, and the Research Corporation of the University of Hawai‘i. We also thank the reviewers for helpful comments on an earlier draft of this manuscript.

Additional Information and Declarations

Competing Interests

Author Contributions

Data Availability

The authors declare there are no competing interests.

Brian J. Neilson conceived and designed the experiments, performed the experiments, analyzed the data, contributed reagents/materials/analysis tools, prepared figures and/or tables, authored or reviewed drafts of the paper, approved the final draft, wrote manuscript, analyzed data, performed and collected the data.

Christopher B. Wall analyzed the data, contributed reagents/materials/analysis tools, prepared figures and/or tables, authored or reviewed drafts of the paper, approved the final draft, wrote the manuscript and performed the analysis.

Frank T. Mancini conceived and designed the experiments, performed the experiments, contributed reagents/materials/analysis tools, authored or reviewed drafts of the paper, approved the final draft, developed the study design, oversaw data collection and management, edited manuscript.

Catherine A. Gewecke conceived and designed the experiments, performed the experiments, contributed reagents/materials/analysis tools, authored or reviewed drafts of the paper, approved the final draft, developed study design, edited the manuscript and collected data.

The following information was supplied regarding data availability:

Neilson, Brian J, Wall, Christopher B, Mancini, Frank T, & Gewecke, Cathy A. (2018). Herbivore biocontrol and manual removal successfully reduce invasive macroalgae on coral reefs. http://doi.org/10.5281/zenodo.1285551.

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
