# Peer review of "Herbivore biocontrol and manual removal successfully reduce invasive macroalgae on coral reefs"

_PeerJ, doi:10.7717/peerj.5332_

## Round 0.1 · original submission · Major Revisions

I have heard back from two reviewers. While both had positive things to say about your work, each had slightly different takes on it. One reviewer offered many editorial revisions and otherwise felt only minor changes would be needed, while the other reviewer feels there is a disconnect between the title and the results/outcome. After looking over the submission myself, I agree with both reviewers. Asides from the editorial and minor comments of reviewer 2, I do agree with reviewer 1's overall assessment. Thus, although no complete restructuring or reanalyses are needed, the revisions needed will likely be more than "minor", and my decision is that (somewhat) major revisions are needed. I look forward to seeing your revised version.

Reviewer 1 ·

Basic reporting

Reporting is clear with sufficient background and references with good statical design, analysis and figures.

Experimental design

Good experimental design

Validity of the findings

As authors themselves write in the discussion, the method is not effective to be recommend for a wide scale management of macro algae problems in the reefs.
Research is good, method is sound, but restricted in its applicability

Additional comments

Authors Neilson et al work related to the control of macroalgae using biological and physical methods is an interesting research carried out in the filed in Coconut Island.
I agree with the concept of the authors that there needs to be some method to control algal outbreak in reefs to help keep balance between algal and coral growths.

Removal or addition of biological organisms to control or prevent something is not new. It all comes to how effective it is in terms of time, manpower, amount spent and final results.
I feel that authors have contrasting ideas when you look at the title-abstract and later part of discussion. It seems to be an effective method when you read only title and abstract, however discussion tells a different story. This is kind of misleading.

Yes, there was a substantial change in the algal cover over time compared to control, but that did not change anything in terms of coral cover? Also on an ecological time scale to see substantial shift, this work is not long-term enough.
As a technique combining physical and biological controls simultaneously, this work is a good example.
Having said that, as authors themselves point out, there are many problems.
The reduction in the algal cover you see may be short-term, they could come back, once physical removal and urchins are stopped.
I do not see any substantial change in coral cover between 2 treatments. So was the presence of macro algae really effecting the coral in treatment area?

It is nice this worked for Coconut Island reef, but it might not be the same if some one try to use this method elsewhere.
There could easily be sea urchin outbreaks. High costs for a long-term period might not be feasible for many management people. Since, this has to be continued as long as macro algae exist in the area.

I think this the scope of this work in terms of its use is not much and limited.
I am not convinced how this can be an effective management tool and can be recommended to managers?

·

Basic reporting

This MS describes a trial for controlling introduced seaweeds on coral reef in Hawai’i, using a combination of manual seaweed removal with subsequent suppression of re-establishment by deploying hatchery-reared grazers (the sea urchin Tripneustes gratilla).

While simple in its design and with only two replicates (due to logistical constraints) the study is a robust demonstration of the effectiveness of the tested control method. The results (including the cost estimates) are very relevant for reef-management and publication is recommended.

Experimental design

The experimental design is fit for purpose, the explanation why a fully orthogonal design was not chosen is sufficient and make sense.

The statistical analysis is appropriate for the simple experimental design.

Validity of the findings

The results are clearly presented, backed up by appropriate figures and tables, and sufficiently discussed.

The only part of the MS that is a bit weak is the discussion about risk and potential side effects of the tested combined control method (removal of macroalgae and increased grazer densities). This could be better structured and more detail (plus references) added.

The efficiency of macroalgae removal is described and briefly discussed but there is little mention of potential side effects, e.g. damage to corals, disruption of reef-associated fauna, and removal of native macroalgae. This should be discussed in more detail.

In line 437-443 the risk of deployment of hatchery raised urchins is discussed. It is interesting to note that the native macroalgae where not affected by the treatment- this should be discussed, eg why were they not consumed by the urchins? Also, is there potential to introduce diseases to other echinoderms? How was this controlled (eg hatchery protocols). Is there competition due to the higher T. gratilla densities with other invertebrates?

Additional comments

Minor comments by line numbers – mix of editorial suggestions and content queries that should be addressed in a revision:
32-34 consider rephrasing
34 add macroalgae (seaweed) before “mariculture”
35 Add “commercial” before Macroalgae
38 suggest to change to “often cultivating non-native or domesticated species of the genera…”
38 genera not generas
40 “opportunities” instead of “incentives”; ad “can” before “offer”
41 “viable” instead of “lucrative”; omit “struggling”
43 Hewitt
44-45 select more suitable references, ie those where reef decline is related to macroalgal cover (there are plenty)
45-48 sloppy wording. Suggest being specific to macroalgal growth being an issue, which you do in the next para. Alternatively reword to describe here the current coral reef “crisis” with many reefs being affected by anthropogenic pressures (most recently the global coral bleaching events), which makes it necessary for reef managers to consider all option to reduce manageable pressures (such as the one you are investigating in your study).
58-60 Reword to make clear that phase shift are a general problem (especially in eutrophied conditions) but that invasive macroalgae add another dimension (as they generally have been selected for traits like fast grow, easy and frequent reproduction). Perhaps add some newer references on phase shifts, e.g.
 Mumby, P.J., Steneck, R.S., Adjeroud, M., Arnold, S.N., 2015. High resilience masks underlying sensitivity to algal phase shifts of Pacific coral reefs. Oikos.
 Dell, C.L.A., Longo, G.O., Hay, M.E., 2016. Positive Feedbacks Enhance Macroalgal Resilience on Degraded Coral Reefs. PLoS ONE 11, e0155049.
 van de Leemput, I.A., Hughes, T.P., van Nes, E.H., Scheffer, M., 2016. Multiple feedbacks and the prevalence of alternate stable states on coral reefs. Coral Reefs, 1-9.
66 “management objective” instead of “response goal”
72 missing “in” before “water”?
84 perhaps “ability to produce and deploy adequate densities of biocontrol grazers”
92 vagility?- do you mean “mobility”?; maricultured?
155 DNLR, write out in full when first tam used
160 “rehabilitation” instead of “restoration”?
165 suggest: “was yet to be tested at a management-relevant, reef-wide scale.”
168 “using” instead of “following”
171 no dash
192 island-like
207 add reference for description of “super sucker” removal technique
215 “branches” instead of “fingers”
225 “manually deployed” or “released” instead of “hand placed”?
229 “deployed” instead of “outplanted”
248 omit “binned”
253 “turf algae cover”- perhaps add definition of turf in your study (eg less than 10 mm height or something)
256 add “initially”
286 just use “vs. control”
288 delete patch reef here
296 what transformations?
301 xxx?
307 delete “biological”
308 hard corals? Not reef…
313 “ had similar cover of each….”
327 I assume this refers to the baseline surveys? If so, what are T. gratilla pre-control at the treatment reefs? Perhaps this information could be better included in previous section.
333 add “cover” after macroalgae, use “different” instead of “affected”
341 add “was”
357 “components” instead of “portions”- deployment of urchins happened only once? No repeat deployment was required?
363 “techniques” instead of “acrions”
369 “a” instead on “the”; add “cover”
379 suggest” reducing the risk of unintentional dispersal”
385 add an estimate of time to the cost estimate results
386 how many and how is this captured in the cost estimate? Did you have a threshold when you decide to deploy additional urchins?
387 does that mean that after full treatment establishment no further urchin re-stocking was required?
403-411 presumably the juvenile urchins grew to adult size during your study. What does this mean for grazing efficiency?
447-448 why would that be? Is that the only explanation for the observation that macroalgae cover also decreased on control reefs?
498 reword, cost does not automatically reflect value
502-504 but the risks of unwanted side-effects may also increase….

---

## Round 0.2 · accepted · Accept

I have read over the manuscript and your responses and revisions, and judge that you have answered all concerns very well; the reviewer who examined this version also states the same. Thus, I am more than happy to move this into production. I look forward to seeing the published version!

# Reviewer 1 ·

Basic reporting

Reporting is clear and professional English used throughout with proper citations and background information.

Experimental design

Well defined questions with good experimental design

Validity of the findings

no comment

Additional comments

The authors have revised previous version of their manuscript and the present version is good and clear. I don't have any more comments with respect, all the comments and suggestions for the previous version of the manuscript has been taken care.
I am happy to see this get through and published.